# Effect of Nano-MgO Particles Doping on Breakdown Characteristics of Polypropylene

**Guang Yu, Yujia Cheng \* and Zhicheng Wu**

Mechanical and Electrical Engineering Institute, University of Electronic Science and Technology of China, Zhongshan Institute, Zhongshan 528400, China; yuguang@hrbust.edu.cn (G.Y.); a26165134@163.com (Z.W.)

\* Correspondence: chengyujia@hrbust.edu.cn; Tel./Fax: +86-760-8826-9835

**Abstract:** In this article, the nano-MgO particles were used as inorganic fillers, and polypropylene (PP) polymer was used as a matrix. The nano-MgO/PP composites were prepared by double melt blending. Using the polarization microscope (PLM) test and hot-stage microscope test, the crystalline morphology of PP and nano-MgO/PP with different mass fraction were observed. Using the differential scanning calorimetry (DSC) test, the parameters and crystallinity in the process of isothermal crystallization could be obtained. Additionally, the samples of pure PP and nano-MgO/PP composites were dealt with using a breakdown test and a dielectric frequency spectra test. From the experimental results, nano-MgO particle doping decreased the samples' crystal size, and the crystalline structure was converted from large spherulites to fascicled crystallization. Additionally, the crystallization rate became fast and crystallinity increased. According to the breakdown test, the nano-MgO particle doping made the composites form small, dense spherulites. The breakdown developed through a longer path, so the composites' breakdown strength rose greatly. When the mass fraction of nano-MgO particles was 3%, the shape parameter of the composites' Weibull distribution β was larger, which illustrated that the nano-MgO particles were dispersed uniformly in the PP matrix. According to the dielectric frequency spectra test, the dielectric constant of different nanocomposites were all lower than which of pure PP, but the loss angle tangent values were all higher than which of pure PP.

**Keywords:** PP; nano-MgO; dielectric properties; crystalline morphology

## 1. Introduction

With the advancement of technology, the development trend of power transfer equipment to large-capacity and high-voltage and the demand for power system reliability have become higher and higher. In order to ensure the steady operation of a power system, the electrical equipment insulation strength and service life must be improved. Normally, cross-linked polyethylene (XLPE) is used for the cable insulation material [1–3], but XLPE has several drawbacks, such as being non-recyclable and polluting the environment. The thermal properties of polypropylene (PP) are better than those of Polythene (PE). Additionally, the chemical resistant property and high temperature performance of PP are excellent, so PP is widely used as the insulation material in power capacitors. While PP works long-term in an environment of high frequency and high field, the electrical treeing spreads easily, which leads to insulation breakdown. Additionally, the mechanical tenacity is poor. As one kind of cable insulation material, the application of PP in high-voltage insulation is restricted by its flaws. With the development of nanotechnology, new methods of polymer modification have emerged. Because nanoparticles possess a large specific surface area, lots of interfaces have been formed by the combination of nanoparticles and polymer. By effectively regulating and controlling these interfaces, the composites' macroscopic properties can be affected [4]. Montanari and Tanaka

carried out a series of studies based on the analysis of montmorillonite (MMT) and its dispersion as an inorganic dispersion phase. In this research, the dielectric properties of nanocomposites were significantly improved compared with which of common composites. This was because the new interface structures were introduced by nanodielectrics [5,6]. This was consistent with the study results of Dissado and Forthergill [7–10]. From the correlation research, nanoparticle doping could improve the dielectric properties of polymers effectively [11–14]. Therefore, a number of metal oxides were added into the polymers such as low-density polyethylene, epoxy resin and polypropylene. With this nanoparticle doping, the matrix corona resistance and breakdown characteristics were significantly improved [15,16]. The formation of electrical treeing was inhibited [17]. Additionally, the space charge distribution [18,19], the polymers' crystalline morphology [20,21], the mechanical properties and flame resistance were all improved [22]. Throughout the former studies on nanocomposites' dielectric properties, the research focus mainly concentrated on three aspects: (1) The inorganic nanofiller changed the polymers' crystalline morphology. The conductive path was prolonged, and the polymer breakdown characteristics were improved. (2) The inorganic nanofiller introduced large amounts of interface structures into the polymers, which possessed a certain scattering effect to the carrier. The polymers' dielectric properties were improved. (3) The inorganic nanofiller introduced deep traps into the polymers. The carrier could be captured and struggled to move freely. Therefore, the mobility of the carrier inside the dielectric decreased. However, in all these studies, only a single experimental phenomenon was analyzed. Therefore, it requires a combination of material structures to explore the effect of nanoparticle doping on polymers' dielectric properties.

PP was one kind of high polymer, in which the crystalline phase and non-crystalline phase coexisted. The polymer dielectric properties were affected by its crystallinity, crystalline morphology and spherulite size. Therefore, the nucleating agent or the particles possessing heterogeneous nucleation effects, could be doped into PP, due to which which polymer microstructures such as crystalline morphology and crystal size would change. Then some material properties could be improved. Recently, a series of research works were carried out on PP homopolymer, PP copolymer and PP nanocomposites, which attempted to improve the electrical properties of PP [23,24]. But little of this literature was about the effect of nano-MgO particle doping on crystalline morphology and the breakdown characteristics of PP. In order to improve the polymers' crystalline morphology, breakdown properties and dielectric properties, the following work, based on the current study situation in nanocomposite dielectrics, was carried out. Firstly, the nano-MgO/PP, with different particle mass fractions, was prepared by melt blending. Then, according to the polarization microscope (PLM) test, differential scanning calorimetry (DSC) test, breakdown test and dielectric frequency spectra test, the effect of nanoparticle doping on the crystalline morphology and dielectric properties of PP was explored. According to the improvement of the composites' dielectric properties, experimental instruction and a theoretical basis is supplied for the development of nanocomposites with high breakdown field strengths. Thus, the polypropylene material can be more extensively used in the field of high-voltage insulation.

## 2. Experimental Method and Samples Preparation

### 2.1. Experimental Materials and Instruments

Experimental materials: polypropylene (the model was T30s; the producer was Fushun Petrochemical Company, Fushun, China; the CAS number was 9003-07-0); MgO (the producer was Beijing Deke Daojin Science and Technology Co., Ltd, Beijing, China; the CAS number was 1309-48-4); the antioxidant 1010 (the producer was North China Special Chemicals Co., Ltd, Tianjin, China).

Experimental instruments: torque rheometer (the model was RM-200; the producer was Haster Technology Development Co., Ltd, Harbin, China); plate vulcanizing press machine (the model was XLB 24-D; the producer was Huzhou Xingli Rubber Machinery Manufacturing Co., Ltd, Huzhou, China); polarization microscope (the model was LeicaDM2500; The producer was Leica Microsystems Co., Ltd, Wetzlar, Germany); differential scanning calorimeter (the model was DSC-1;

the producer was Mettler Toledo, Columbus, OH, USA); AC high-voltage experimental console (the model was JG-5; the producer was Shanghai Pujing Electrical Co., Ltd, Shanghai, China); broadband dielectric spectra analyzer (the model was Alpha-A; the producer was Novocontrol Technologies, Montabaur, Germany, and the test frequency was $10^1$ to $10^5$ Hz).

*2.2. Samples Preparation*

In this experiment, double melt blending was used for the composites' preparation. During the preparation process, the nano-MgO and PP particles were placed into torque rheometer for melt blending directly. The process of composite preparation by double melt blending is shown in Figure 1 [25].

The specific steps were as follows:

Firstly, nano-MgO particles and PP were mixed in proportions of 1:10. After melt blending by torque rheometer (Haster Technology Development Co., Ltd, Harbin, China), a master batch of MgO/PP composite was prepared.

Secondly, this master batch was diluted into the PP matrix with appropriate proportions. The mixture was prepared with melt blending by torque rheometer again, then nano-MgO/PP with 1%, 3% and 5% particle mass fractions was prepared completely.

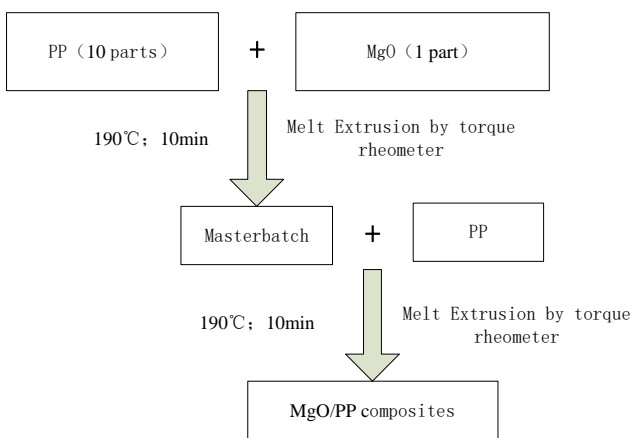

**Figure 1.** The flow chart of melt blending polymerization.

All the composites were taken out from the torque rheometer and prepared with hot pressing by plate vulcanizing press machine. The process parameters were set at: 190 °C, 10 MPa and 10 min. After hot pressing, these samples were pressed to 100 μm thickness. Finally, these samples were cooled down to room temperature slowly and placed into a drying tank for use.

*2.3. Microstructure Characteristics and Macroscopic Test*

2.3.1. PLM Test

In this experiment, the model of PLM was LeicaNM2500, which was produced by Shanghai Leica Microsystems Trade Co. Ltd, Shanghai, China. In order to observe the crystalline morphology of PP and MgO/PP composites, the samples must be prepared with surface corrosion first. The corrosive agent was concentrated sulfuric acid and potassium permanganate with 5% content [26]. The concentration of sulfuric acid was 98%. At room temperature, the samples of PP and MgO/PP composites were corroded for 5–6 h, and the corrosion solution was stirred every half an hour. After the corrosion, the samples were taken from the corrosion solution and rinsed with clean water. Then these samples were treated with ultrasonic cleaning by ultrasonic cleaners for 15 min and dried under certain conditions. After that, the samples for PLM test were prepared completely.

The crystalline morphology of different samples could be observed by PLM. Firstly, the hot-stage temperature rose to 200 °C. In order to eliminate the thermal history residue effect, the temperature was retained for 10 min then cooled slowly until the appearance of polymer

crystallization. From the PLM test, when the temperature dropped to 135 °C, the crystal nucleus could be observed from PLM. At this time, the crystallization rate was fast and crystallized sufficiently. Finally, the PLM patterns of test samples were obtained.

2.3.2. DSC Test

In this experiment, the model of DSC was DSC-1, which was produced by Mettler Toledo Group Co., Ltd. The isothermal crystallization process of PP and nano-MgO/PP composites with different particle mass fractions was explored using the DSC (Mettler Toledo DSC-1) test. Under nitrogen protection with a 150 mL·min$^{-1}$ flow rate, the samples' temperature rose to 190 °C in 10 °C·min$^{-1}$, from which the samples' thermal history could be eliminated. Then the samples' temperature was dropped down to 25 °C with 10 °C·min$^{-1}$ cooling rates. The enthalpy changes in the cooling process were recorded, from which the crystallization temperature $T_c$ and the width of exothermic crystallization peak $\Delta T_c$ could be calculated. When the crystallization was complete, the samples' temperature rose to 190 °C in 10 °C·min$^{-1}$ again. The enthalpy changes in the heating process were recorded, from which the melting temperature $T_m$ and samples crystallinity $W_c$ could be calculated.

From DSC test, the composites crystallinity $W_c$ could be calculated by Equation (1).

$$W_c = \frac{\Delta H_m}{(1-\omega)H_0} \times 100\% \tag{1}$$

In Equation (1), $\Delta H_m$ is the melting enthalpy. $H_0$ is the melting enthalpy under full crystallization, which was 201 J/g for PP [27]. ω is the MgO mass fraction in composite.

2.3.3. Breakdown Test

A power frequency AC system was used for the breakdown field strength test. This system boosted at a speed of 1 kV/s until the materials reached breakdown, and the breakdown field strength $U$ was recorded. Then, the thickness of breakdown points d was measured. According to the formula $E = U/d$, the breakdown field strength of different samples $E$ could be calculated. Two breakdown points voltage per sample were tested, and the test data were analyzed by MINITAB (version 17). The samples' shape parameter β and breakdown field strength $E_0$ under Weibull distribution were obtained, from which the Weibull distribution curve of different samples could be drawn. The Weibull distribution expression for β and $E_0$ is shown in Equation (2).

$$P(E) = 1 - \exp(-(\frac{E}{E_0})^\beta) \tag{2}$$

In Equation (2), $P$ is the samples' breakdown probability; $E$ is the samples' breakdown field strength; β is the shape parameter of data dispersion; $E_0$ is the scale parameter of electric field strength when $P$ is 63.2%.

The broadband dielectric spectra analyzer was used for a dielectric frequency spectra test, and the test range was set to 1–10$^4$ Hz. The samples' thickness was 100 μm and the diameter was 40 mm. Under vacuum conditions, the two electrode systems in these samples were evaporated, and the aluminum electrode diameter was 25 mm. Before the dielectric frequency spectra test, these samples were placed into an 80 °C oven. After short circuit treatment for 24 h, the effect of samples' moisture and residual charge on this test was eliminated.

## 3. Experimental Result and Analysis

### 3.1. PLM Characterization of MgO/PP Composites

The PLM patterns of pure PP and different MgO/PP samples are shown in Figure 2.

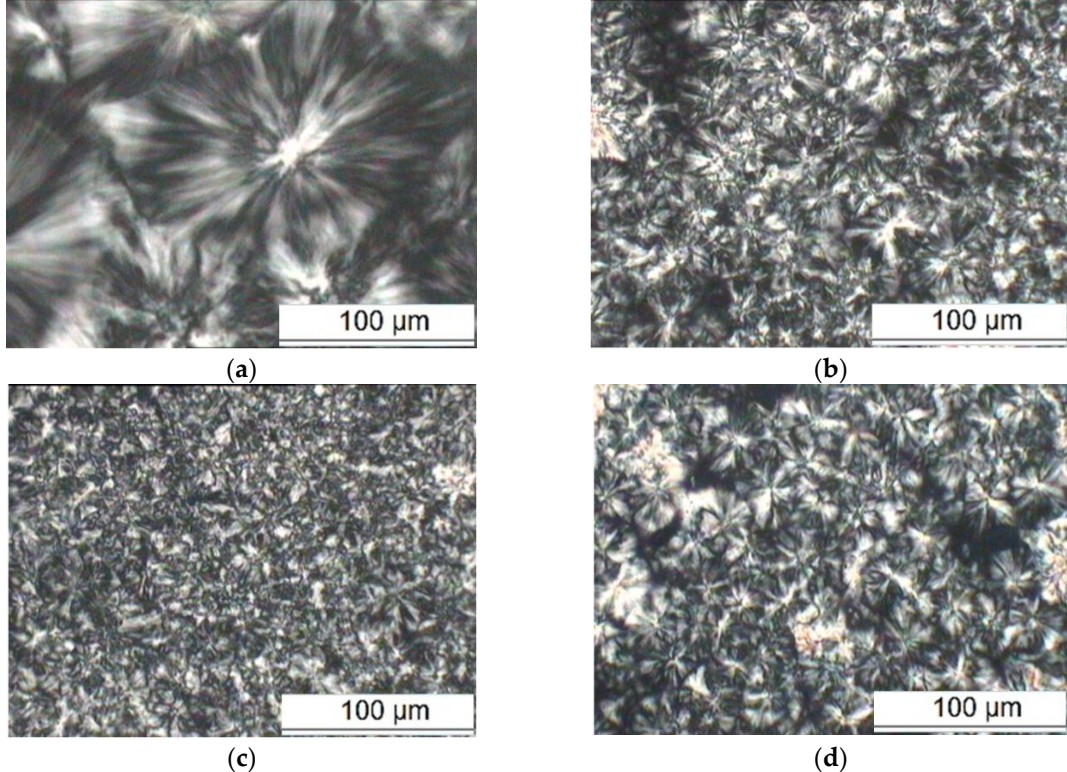

**Figure 2.** PLM patterns of nano-PP/MgO composites: (**a**) PP; (**b**) 1% nano-MgO/PP; (**c**) 3% nano-MgO/PP; (**d**) 5% nano-MgO/PP.

In Figure 2, the Maltese cross extinction pattern can be observed clearly from PLM patterns of PP sample. The crystal size was larger and reached about 100 μm. The grains were arranged uniformly and orderly. The grain interface of the crystalline region and non-crystalline region are visibly displayed. Figure 2b–d shows the PLM patterns of different nano-MgO composites. With nano-MgO particle doping, the grain size reduced significantly. When the particles' size was reduced to 30 μm, the spherulites' quantity increased per unit area. The interface of crystalline region and non-crystalline region tended to be vague. The crystalline part took on the fasciculation and did not form complete spherulites. With the trend of crystal size reduction, the regions between fascicled crystallization were enlarged. This illustrated that the MgO particles acted as the anisotropic nucleating agent in PP crystallization. A large amount of PP molecular chains grew around a more crystal nucleus, which refined the crystal structure of PP. As the MgO content continued to increase, the Maltese cross extinction pattern was increasingly blurred. This was because too many MgO particles dispersed in the PP matrix, which hindered the order motion of PP molecular chains. The PP molecular chains were frozen before they could self-unwrap in the cooling crystallization, so the molecular chains could not enter into the spherulite structure. Therefore, in the process of PP crystallization, the crystal nucleus grew within a confined space. The spherulites' size decreased and the boundary between spherulites was blurred, which reduced PP crystallinity [28].

### 3.2. DSC Characterization of MgO/PP Composites

The DSC heating and cooling process curves of PP and different nano-MgO/PP composites are shown in Figure 3. In the process of isothermal crystallization, the heating and cooling rate were both 10 °C·min⁻¹. Additionally, the parameters of isothermal crystallization and melting practice of different samples are shown in Table 1.

**Table 1.** Isothermal crystallization and melting process parameters of different nano-MgO/PP and PP specimens.

| Sample | $T_m$ (°C) | $\Delta T_c$ (°C) | $T_c$ (°C) | $X_c$ (%) |
|--------|-----------|------------------|-----------|-----------|
| PP | 173.8 | 15.8 | 118.3 | 46.9 |
| 1% | 174.6 | 14.3 | 122.0 | 47.9 |
| 3% | 176.2 | 12.7 | 125.3 | 49.2 |
| 5% | 174.2 | 13.0 | 120.1 | 48.3 |

Note: $T_m$—The melting peak temperature; $T_c$—The crystallization peak temperature; $\Delta T_c$—The width of Exothermic crystallization peak; $X_c$—crystallinity.

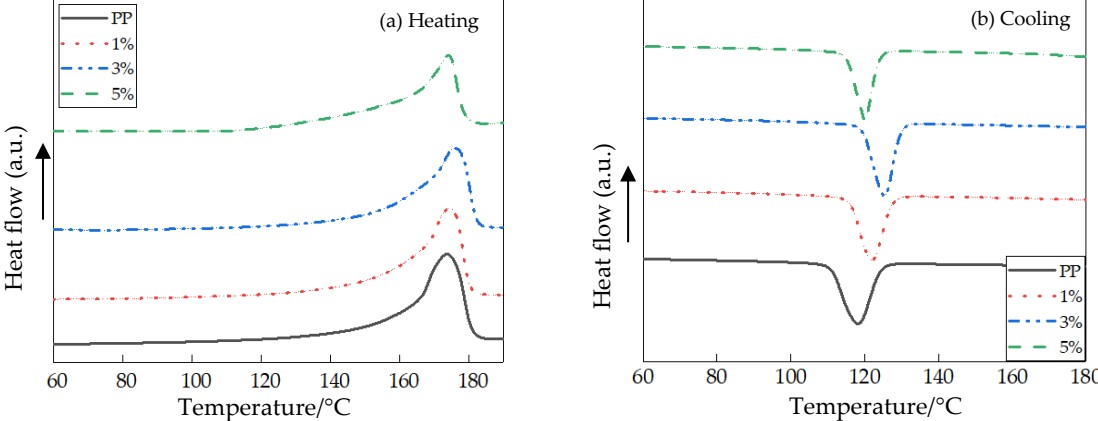

**Figure 3.** DSC heating and cooling process curves of different samples: (**a**) Heating process curve; (**b**) Cooling process curve.

Combining the experimental results of Figure 3 and Table 1, the crystallization peak temperatures $T_c$ of different MgO/PP composites were all higher than those of pure PP at the same cooling rates. This illustrated that the samples crystallized at high temperature with nano-MgO particle doping. During DSC orientation, the lower the temperature of the samples' crystallization peaks, the lower the crystallization rates of polymer. The temperature, in which crystal formation rate was highest, was selected as $T_c$. When the mass fraction of nano-MgO particles was 3%, the $T_c$ of MgO/PP composite was the highest, 125.3. At this time, the samples' crystallinity was also the highest, 49.2. The crystallization rate order of the four samples was as follows: PP < 5% < 1% < 3%. From the test results, MgO particle doping improved the crystal formation rate of PP, and the width of the exothermic crystallization peak $\Delta T_c$ of nanocomposites was less than that of pure PP. This illustrates that the effects of different MgO particle sizes on crystal formation rate of PP were different. The crystallinity values of nano-MgO-composites with 1%, 3% and 5% particle mass fractions were 2.1%, 4.9% and 2.9% higher than that of pure PP, respectively.

### 3.3. Breakdown Test of MgO/PP Composites

The AC breakdown test results of different nano-MgO/PP composites are shown in Figure 4. In this experiment, the breakdown test result was characterized by a Weibull distribution. Compared with common data plotting, the Weibull distribution can evaluate the insulating material breakdown action under an AC electric field effectively. This reflects the probability of samples being broken down under a certain electric field strength; in other words, it reflects the material failing probability at a certain time.

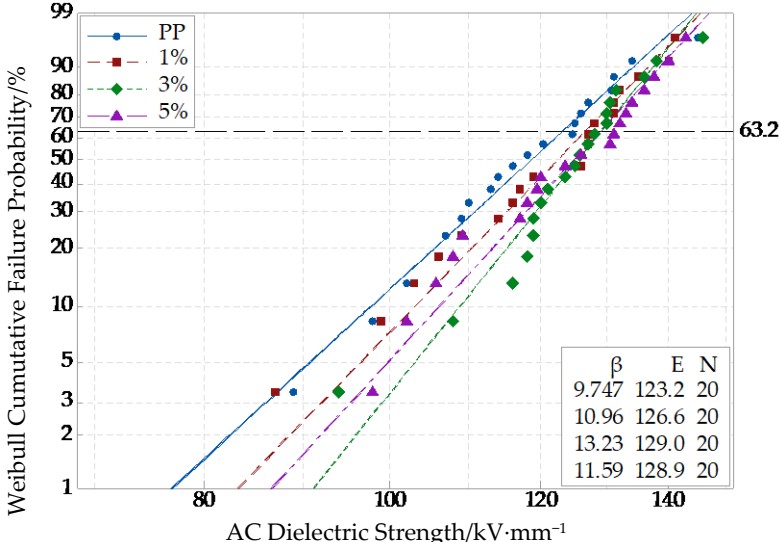

**Figure 4.** Weibull distribution curve of AC breakdown strength of various samples.

　　Shown in Figure 4, the breakdown field strength values of different nano-MgO composites were all higher than that of pure PP. With the increasing of nano-MgO particle mass fraction, the breakdown field strength first increased and then decreased. When the mass fractions of nano-MgO particle doping were 1%, 3% and 5%, the breakdown field strengths of the nano-MgO/PP composites were 2.76%, 4.7% and 4.6% higher than that of pure PP. Compared with the common melt blending, the composites' breakdown field strengths when prepared by double melt blending were higher, and the nanoparticles' dispersion was better in matrix. This was because the nanoparticles' dispersion was better, and the distance between nanoparticles in the matrix resin was greater. Additionally, the interface structure was looser. Therefore, the electrons migrated more easily, which was good for conductive path formation. In addition, the nanoparticles dispersed uniformly in the polymer matrix. The scattering effect of dipole on electrons was strengthened [29,30], so the breakdown field strengths of MgO/PP composites were higher than that of pure PP. With an increase in nano-MgO-particles, the distance between nanoparticles decreased, to form a close interface structure. The electronic transition had to overcome a higher barrier, and the scattering effect of the interface on the electrons was strengthened. Then, the electrons' directional migration was restrained, which reduced the free electrons' quantity in the interface region. Therefore, the breakdown field strength of composites increased. At this time, the nanoparticles doping promoted the deep traps formation. Additionally, the traps' dispersion rate was improved. The probability of free electrons being captured increased, which improved the breakdown field strength of polymer effectively. When the particles' mass fractions continued to increase, the nanoparticles agglomeration appeared in the matrix, and the nanoscale effect weakened. Therefore, the breakdown field strength of composites decreased slightly. Additionally, more electric charges accumulated in interface regions, which caused electric field distortion. For this reason, the breakdown field strength of composites decreased as well.

　　The shape parameter and breakdown field strengths of different samples for Weibull distribution are shown in Figure 5. The scale parameter $E$ reflects the samples breakdown field strength, and the shape parameter $\beta$ reflects the dispersion of breakdown voltage. In the matrix materials, a larger $\beta$ indicates a better dispersion.

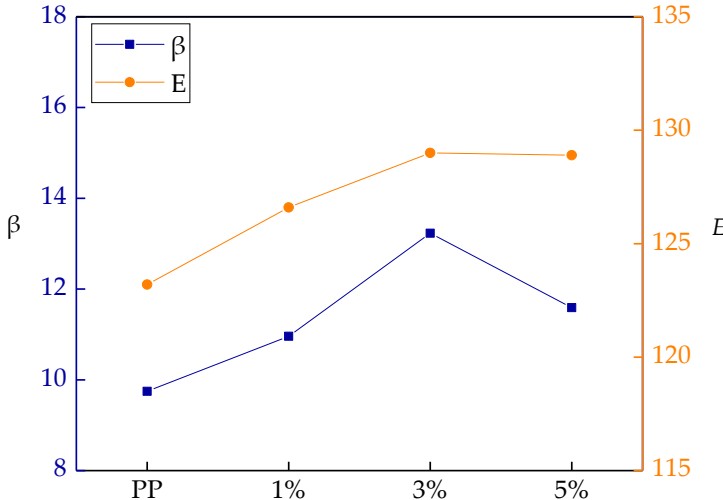

**Figure 5.** Weibull parameters of AC breakdown strength of various samples.

As shown in Figure 5, with different mass fractions of nano-MgO particle doping, the particles' dispersions in the matrix were different. When the particles' mass fraction was 3%, the composite shape parameter β was the largest. This illustrated that the nano-MgO particles dispersed uniformly in matrix resin. The different interface microdomains were formed by the different mass fractions of particle doping, so the interphase combining degrees were also different. From the test results, when the mass fraction was 3%, a close interface structure was formed in the composite. In this interface, more carriers were captured, which reduced the carrier concentrations of composites. Therefore, the breakdown field strength increased [31].

### 3.4. Dielectric Frequency Spectra Test of MgO/PP Composite

The relations between the relative dielectric constant $\varepsilon_r$ and the frequency of different samples are shown in Figure 6. The relative dielectric constant showed a downward trend with increasing frequency. The change rules of nanocomposites and pure PP were identical. This illustrated that the polarization establishment processes of nanocomposites and pure PP were also identical. They all accorded with the general law of dielectric relative dielectric constant. In low applied frequency, the dielectric constant was basically unchanged. As the frequency increased, the dielectric constant decreased gradually. This was because nanoparticle doping formed an interface structure in the polymer matrix, which inhibited the movement of PP macromolecule chains. The polarization establishment process suffered serious obstacles, so the dielectric constant was lower [32]. In the low-frequency region $\omega\tau \leq 1$, $\varepsilon_r \rightarrow \varepsilon_s$ was established promptly in both instantaneous polarization and relaxation polarization. When $\omega\tau$ approached 1, the macromolecule chains of PP could not keep up with the gradually increasing change of applied electric field, so the relaxation polarization could not be established. The relaxation polarization made less of a contribution to the dielectric constant of composite, which reduced the $\varepsilon_r$ of MgO/PP composites. As the applied electric field continued to increase, the dipolar polarization of MgO/PP composites could not keep up with the changing of the applied electric field. At this time, the displacement polarization played the major role in the polarization process. Hence, the dielectric constant of composites showed a downward trend until it dropped down to a relatively stable value. From the test results, with the frequency increasing, the downward trend of composites' dielectric constants was obvious and finally stabilized. The dielectric constant of pure PP was 2.5 and higher than that of all the composites. Among the composites, when the mass fraction of nano-MgO particles was 1%, the dielectric constant was the highest, 2.43. This was because nano-MgO particle doping produced physical crosslinking points in the movement of PP macromolecule chains, which exerted the small size effect and surface effect on nanoparticles. Additionally, more interface active regions were introduced by the inorganic particle doping. A closer interface structure was formed between nanoparticles and polymer through the covalent bond.

In summary, nano-MgO-particle doping could limit the movement of polymer macromolecule chains effectively, and the dielectric constants $\varepsilon_r$ of nanocomposites were lower than that of pure PP.

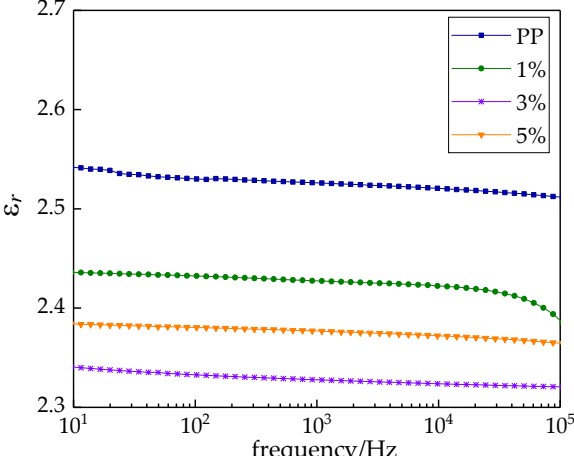

**Figure 6.** Frequency dependences of relative permittivity of MgO/PP composite.

The relations between the loss angle tangent tanδ and frequencies of different samples are shown in Figure 7. As the frequency increased, the loss angle tangent value decreased. For all test frequencies, the loss angle tangent values of composites were all higher than which of pure PP. As the applied frequency increased, the loss angle tangent values of nano-MgO/PP composites showed a downward trend until they dropped down to a relatively stable value. This was because, in low-frequency regions, $\omega\tau$ was less than or equal to 1, so the loss angle tangent values of composites were mainly produced by conductive loss. When $\omega\tau$ approached 0, the value of tanδ approached infinity. In summary, conductive loss played a major role in low-frequency regions, and tanδ was larger. But in high-frequency regions, the relaxation polarization was hard to establish. At this time, the loss was mainly produced by displacement polarization. The value of tanδ decreased gradually then dropped down to a stable value.

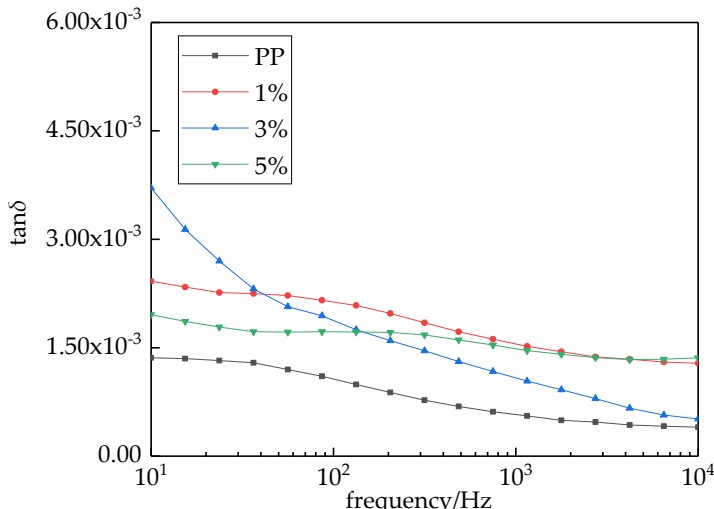

**Figure 7.** Frequency dependences of dissipation factor of MgO/PP composite.

The value of tanδ represents the loss degree directly. The dielectric loss only depended on the material's characteristics, and not on its size and shape. A great dielectric loss led to dielectric heating, in which the dielectric properties were destroyed [33]. When the mass fraction of nano-MgO particles was 3%, strong interface bonding was formed between inorganic phases and organic phases of composites. The interaction force was strong. When the polarization happened, the dipole orientation

polarization was limited by the interface region. Therefore, the dielectric constant was small and the dielectric loss was high.

## 4. Conclusions

Using melt blending, nano-MgO/PP composites with 1%, 3% and 5% particle mass fractions were prepared. The crystalline morphology and crystalline process of different composites were observed by PLM and DSC test. Additionally, the effect of nano-MgO particle doping on the insulation characteristics of PP could be explored by breakdown test and dielectric frequency spectra test. The conclusions were as follows:

- The crystalline morphology of different samples was observed by PLM. Nano-MgO particle doping acted as an anisotropic nucleating agent, which changed the original crystalline structure of PP. With moderate nano-MgO particle doping, the particles' dispersion in PP matrix was better. The crystalline structure of composites was uniform and close. Additionally, the crystal size reduced, and the crystal quantity per unit space increased.
- Nano-MgO particle doping changed the ordered arrangement degree of molecular chains in PP crystallization. The PP crystallization was converted from complete large spherulitic crystal structure to a fasciculation crystal structure. The Maltese cross extinction patterns were increasingly blurred in PLM patterns. Because of the crystal nucleation centers increasing and the crystalline structure changing, the crystallization rate and the crystallinity of nano-MgO/PP composites were improved.
- According to the power frequency AC breakdown test, different mass fractions of nano-MgO particle doping improved the breakdown field strength of PP with varying degrees. Among them, when the nano-MgO particle mass fraction was 3%, the breakdown field strength of the nanocomposites was the highest and 4.7% higher than that of pure PP.
- According to the dielectric frequency spectra test, the dielectric constants of different nanocomposites were lower than that of pure PP. For all test frequencies, the loss angle tangent values of composites were higher than that of pure PP. Among them, when the nano-MgO particles mass fraction was 3%, the dielectric constant of the nano-MgO/PP composite was closest to that of pure PP.

In further research, there may be two continuations of this research topic. One is the effect of micro- and nanoparticle doping on the dielectric properties of PP, the other is research on crystalline morphology and interface microdomain structures of micro-nano-MgO/PP composites.

**Authors Contributions:** Conceptualization, G.Y.; Methodology, Y.C. and G.Y.; Formal Analysis, Y.C.; Investigation Y.C. and Z.W.; Resources, G.Y. and Z.W.; Data Curation, Y.C. and G.Y.; Writing—Original Draft Preparation, G.Y.; Writing—Review and Editing, Y.C. and Z.W.; Supervision, G.Y.; Project Administration, G.Y. All authors have read and agreed to the published version of the manuscript.

**Funding:** This research was aided by the Higher Education and Teaching Innovation Project of University of Electronic Science and Technology of China, Zhongshan Institute, China University under grant No. JY201904.

**Acknowledgments:** Thank Guang Yu for his help.

**Conflicts of Interest:** The authors declare no conflict of interest.

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
