# Peer review of "Effect of Nano-MgO Particles Doping on Breakdown Characteristics of Polypropylene"

_coatings, doi:10.3390/coatings10040312_

Round 1

Reviewer 1 Report

The submitted manuscript is entitled “Effect of Nano-MgO particles doping on breakdown characteristics of PP”.

The authors investigate the effect of nanoparticle doping on the crystalline morphology and dielectric properties of PP. The research topic is interesting.

1. The writing style is not good. However, the article is readable.

2. The authors should provide information about PLM and dielectric equipment (as for DSC).

3. The main problem is the presentation of the analysis of crystallinity. The authors should add DSC thermograms to the manuscript.

4. Unfortunately, the authors did not use any model for the characterization of the crystallization process.

5. In table 1 information for isothermal crystallization is provided.  The authors claim that DSC measurements were carried out with a 10 oC/min scanning rate. How does it relate to the isothermal conditions?! More information is needed.

6. The quantitative analysis of the process is necessary.

Until the authors do not describe the crystallization process in more detail, the article cannot be recommended for publication. 

Reviewer 2 Report

In this manuscript the authors discussed about the possibility to fabricate MgO nano particles/PP composites with enhanced dielectric properties, in particular with higher breakdown field strength. Composites structures have been studied through PLM, DSC and Breakdown tests underlighting the effect of nano MgO on the properties of final composites. The paper should be improved in some parts before the publication on Coatings journal, be based on the following comments:

  • Please, define PLM and DSC acronyms in the abstract;
  • Please, define XLPE, PP and PE acronyms in the text;
  • Introduction section: it seems from what you wrote that you are the first in using MgO nanoparticles to improve the dielectric performances of PP. Actually, you need to cite the paper wrote by Zhou et al, Journal of Applied Polymer Science, Vol. 133 (1), 2016 and critically discuss what are the differences between your work and that paper. Are there some advances in your paper with respect to the work of Zhou et al.?
  • Experimental section: what do you mean with Nano-MgO particles? Are they commercial or you prepared them? Please, specify better in the text;
  • Experimental section: you wrote that PP and MgO are in ratio 1:10. This is in contrast with that reported in Figure 1. Please, amend this error;
  • “Sulphuric Acid and Potassium Permanganate with 5% contents”: what do you mean? Please specify if it is an aqueous solution and what are the molarities or wt.% of two compounds;
  • From Figure 2 it is not possible to understand what are the dimensions of crystal size. Please, correct the figure adding a notch for the size;
  • There are many typos in the text (e.g., page 1: “properties of PP WERE”, 3 times “MALTESE”, in Figure 2 caption: 2d), page 7: 4 times RELAXATION polarization, DIELECTRIC constant). Please, read carefully the manuscript and amend all the typos;
  • Please, adjust the position of the text of the axis in Figures 2, 3, 5 and 6;
  • DSC analysis: how do you explain that the composite with 5% is not in trend with the other ones? Please, critically discuss this dependence in the text;
  • How did you measure dielectric constants? Please, explain better in the text;
  • “In summary, the Nano‐MgO‐particles doping could limit the movement of polymer macromolecule chains effectively, and the dielectric constant εr of Nano‐composites were lower than which of pure PP.” It seems that this behavior is due only to the presence of nanoparticles, thus any type of nanoparticles could have a beneficial effect, despite MgO has a higher dielectric constant with respect PP. How do you explain this? The dielectric constant of MgO has no effect on dielectric properties of the composites? Please, discuss this aspect in the text.

Reviewer 3 Report

In this paper Authors were used the Nano-MgO particles  as inorganic fillers, and polypropylene polymer PP was used as matrix. The Nano-MgO/PolyPropylene composites were prepared by double melting blend. Then, to determine the effect of nanoparticles doping on the crystalline morphology and dielectric properties of PP  the PLM test, DSC test, breakdown test and dielectric frequency spectra test were made. The authors also tried to improve the dielectric properties of the polymer composite. Below I presented some remarks that came to my mind during reading.

Remarks:

1. The acronyms should be avoided in the title of a scientific paper even if they are widely known by the experts in the field. Moreover, in the text of the manuscript, even if an acronym's meaning is widely known, it should be explained the first time when it appears. Consequently, the title of the paper should be revised either by replacing the acronyms (PP) with its meaning or by modifying the title as to avoid the acronym's use.

2. In the Abstract, the Authors should also briefly present the results of the research and their application. In its current state, the abstract is more like conclusions.

3. In my opinion the Introduction must be improved. Introduction should adequately represent the state of knowledge and clearly specify the purpose and motivation of taking up the topic. The area of research must be introduced with details for unfamiliar readers. The Authors should state what has justified using the given method, what is special, unexpected, or different in their approach. In the introduction Authors should discuss the references [4-19] in more detail - show exactly what methods were used, what results were obtained and compare them with each other. Please also organize the introduction section as this order: importance and meaning, previous studies (literature review), the gap between previous studies and present studies, objectives. Especially, the objectives should be clear. I consider that the manuscript under review will benefit if the authors make all of these aspects as clear as possible to the readers.

4. Section 2. Experimental Method and Samples Preparation.

a) Authors should give more details about used, additives/compounds, e.g. producer, CAS number, etc.

b) On what basis were the concentrations of the tested compounds selected? Why were you choosing mix proportion 1:10 and particles mass fraction 1%, 3%, and 5%?

c) A reference to Figure 1 is missing. Describe Figure 1.

d) Authors should also provide basic information about the devices used in the process of sample preparation (model, manufacturer, etc.).

e) Based on what standard was the breakdown test carried out?

5. Values and units should be written separately.

6. Figure 2: Figures a) and b) have the inscription "Diffraction Intensity / a.u.".

7. Figure 3: OX axis and OX axis signature are illegible.

8. Figures 5 and 6: The signature of the OX axis is superimposed on the values on the OX axis.

9. In discussion part of section 3, the Authors should make a comparison between their method from the manuscript and other ones that have been developed and used in the literature for the same purpose. The Authors should also highlight what are the advantages and disadvantages when comparing their devised solution with other solutions from the scientific literature. Such comparison significantly raises the meaning of the presented paper.

10. In the conclusions, it would be useful to add information on further research of the authors related to the continuation of this research topic.

11. References should be prepared in accordance with the Coatings template.

Round 2

Reviewer 1 Report

You should add information about BDS equipment.

1. In the caption for Table 1, you should add information about the experiment type.

2. In Figure 3 or caption for Figure 3, you should add information about the direction of exo process and heating and cooling rate of non-isothermal measurement.

3. Unfortunately, the authors did not provide DSC charts of the isothermal crystallization process (Heating flow vs time; for example in inset or additional graph in Figure 3).

Please consider these procedures. This will make the story easier to follow.

Author Response

Response to Reviewer 1 Comments:

1. In the caption for Table 1, you should add information about the experiment type.

According to the expert advice, the equipment information were added. For example, the experiment instruments and method were introduced in section 2.2.

2. In Figure 3 or caption for Figure 3, you should add information about the direction of exo process and heating and cooling rate of non-isothermal measurement.

The direction of exothermic process was marked by the arrow. The heating and cooling rate during the isothermal test were added in the explanation of figure 3. The temperature rising and falling was in the rate of 10°C·min-1.

3. Unfortunately, the authors did not provide DSC charts of the isothermal crystallization process (Heating flow vs time; for example in inset or additional graph in Figure 3).

The isothermal crystallization DSC curve of different samples were added into the manuscript. Because of the epidemic, my laboratory was sealed off. We regret so much that we could not provide the information about the Heating flow vs time.

Reviewer 2 Report

  1. Experimental section: You must specify also the concentration of Sulphuric acid;
  2. You did not amend the typos that I reported in my previous report: Maltase à Maltese, Relacation à Relaxation, Figure 2: b) 5% Nano‐MgO/PP à d) 5% Nano‐MgO/PP. Please amend all the typos;
  3. You did not answer to point 12 of my previous report, reporting only what you already wrote in the manuscript. How do you explain that a material with a higher dielectric constant (MgO) leads to a composite with lower dielectric constant? If I add TiO2 nanoparticles (with much higher dielectric constant), is the effect the same? MgO has a higher dielectric constant, thus I expect a higher dielectric constant in the composite. You need to explain in the text why the addition of a material with a higher dielectric constant has a contrary effect on composite MgO/PP dielectric constant.

Author Response

Response to Reviewer 2 Comments:

1. Experimental section: You must specify also the concentration of Sulphuric acid;

In experimental section, the concentration of Sulphuric acid was 98%.

2. You did not amend the typos that I reported in my previous report: Maltase à Maltese, Relacation à Relaxation, Figure 2: b) 5% Nano‐MgO/PP à d) 5% Nano‐MgO/PP. Please amend all the typos;

According to the expert advice, all the typos in this manuscript were amend.

3. You did not answer to point 12 of my previous report, reporting only what you already wrote in the manuscript. How do you explain that a material with a higher dielectric constant (MgO) leads to a composite with lower dielectric constant? If I add TiO2 nanoparticles (with much higher dielectric constant), is the effect the same? MgO has a higher dielectric constant, thus I expect a higher dielectric constant in the composite. You need to explain in the text why the addition of a material with a higher dielectric constant has a contrary effect on composite MgO/PP dielectric constant.

Although the dielectric constant of Nano-MgO particles was high, after surface modification, the organic group existed in the surface of Nano-MgO particles. So the Nano-MgO particles were covered by the polar group. At this time, it was not just the interaction between the particles and polymer chains in interface, but the interaction of Nano-MgO particles, surface treating agent and polymer changed the polarization behavior in interface area. Therefore, the dielectric constant of Nano-MgO/PP was lower than which of pure PP. Because the interface adsorption was formed between the resin matrix and nanoparticles, the movement of molecular chains or lattice atoms were limited by the interface between organic phase and inorganic phase. Under the low field strength (1×106V/m), the electrostatic force formed in interface was much higher (about 103 times) than the electric force applied by external electric field. The dipole were difficult to convert to the direction of external electric field. Therefore, the relaxation polarization effect was weakened, and the relative dielectric constant decreased.

Reviewer 3 Report

The comments and corrections made as part of my first review have been addressed. The authors have taken into consideration all my questions and comments. I recommend to accept the new version of the paper.

Author Response

Thank you for your confirmation.